# A global meta-analysis of soil organic carbon in the Anthropocene

Damien Beillouin [1,2] ✉, Marc Corbeels[3,4], Julien Demenois[3,5,6], David Berre[3,7,8], Annie Boyer[9], Abigail Fallot[10,11], Frédéric Feder [12,13] & Rémi Cardinael [3,14,15]

Anthropogenic activities profoundly impact soil organic carbon (SOC), affecting its contribution to ecosystem services such as climate regulation. Here, we conducted a thorough review of the impacts of land-use change, land management, and climate change on SOC. Using second-order meta-analysis, we synthesized findings from 230 first-order meta-analyses comprising over 25,000 primary studies. We show that (i) land conversion for crop production leads to high SOC loss, that can be partially restored through land management practices, particularly by introducing trees and incorporating exogenous carbon in the form of biochar or organic amendments, (ii) land management practices that are implemented in forests generally result in depletion of SOC, and (iii) indirect effects of climate change, such as through wildfires, have a greater impact on SOC than direct climate change effects (e.g., from rising temperatures). The findings of our study provide strong evidence to assist decision-makers in safeguarding SOC stocks and promoting land management practices for SOC restoration. Furthermore, they serve as a crucial research roadmap, identifying areas that require attention to fill the knowledge gaps concerning the factors driving changes in SOC.

Soil organic matter (SOM), mainly composed of carbon, is a critical component of soils[1]. It plays a major role in regulating soil health[2,3] and other ecosystem services such as biodiversity conservation and food production[4]. Moreover, it is a key contributor to the global carbon cycle and climate regulation[5]. The global soil organic carbon (SOC) stocks to 2 m of soil depth are estimated at approximately 2400 Gt C[6], which is three times the amount of carbon in the atmosphere[7]. Small changes in SOC stocks can, therefore, significantly impact atmospheric carbon dioxide ($CO_2$) levels and climate change[8,9].

Achieving global net zero $CO_2$ emissions by 2050 is crucial for limiting global warming to 1.5 °C by the end of the century[10]. This requires avoiding, reducing, and offsetting greenhouse gas (GHG) emissions in all sectors, including the agriculture, forestry, and other land uses (AFOLU) sector[11]. For instance, it is critical to preserve carbon-rich ecosystems like peatlands, old-growth forests, wetlands, and mangroves, which together hold at least 260 Gt of 'irrecoverable' carbon[12]. On the other hand, negative emission technologies can offset excess GHG emissions[10], and natural climate solutions such as SOC restoration can play a crucial role in this process, while also offering additional benefits such as biodiversity conservation[13–15]. SOC preservation and restoration alone can contribute up to 25% of the potential of natural climate solutions, with 40% of this potential coming from SOC preservation and 60% from the restoration of depleted SOC stocks[16].

Numerous factors, hereinafter referred to as 'drivers', directly or indirectly impact SOC levels. Land-use change is a major SOC driver at

[1]CIRAD, UPR HortSys, Montpellier, France. [2]HortSys, Univ Montpellier, CIRAD, Montpellier, France. [3]AIDA, Univ Montpellier, CIRAD, Montpellier, France. [4]IITA, Nairobi, Kenya. [5]CIRAD, UPR AIDA, Turrialba, Costa Rica. [6]CATIE, Centro Agronómico Tropical de Investigación y Enseñanza, Turrialba, Costa Rica. [7]CIRAD, UPR AIDA, Bobo-Dioulasso, Burkina Faso. [8]CIRDES, USPAE, Bobo-Dioulasso, Burkina Faso. [9]CIRAD, DGDRS, DIST, Montpellier, France. [10]CIRAD, UMR SENS, Montpellier, France. [11]SENS, Univ Montpellier, CIRAD, Montpellier, France. [12]CIRAD, UPR Recycle et Risque, Montpellier, France. [13]Recycle et Risque, Univ Montpellier, CIRAD, Montpellier, France. [14]CIRAD, UPR AIDA, Harare, Zimbabwe. [15]Department of Plant Production Sciences and Technologies, University of Zimbabwe, Harare, Zimbabwe. ✉e-mail: damien.beillouin@cirad.fr

the global scale[17,18]. According to Winkler et al. (2021)[19], almost a third of the global land area has undergone land-use change in the last six decades (1960–2019). The conversion of natural ecosystems to agricultural land is estimated to have resulted in a carbon debt of 116 Gt in the top 2 m soil layer[19]. For example, the conversion of primary forest to cropland in the tropics caused a 25–30% loss of SOC stocks[20]. Afforestation of cropland can partially restore these stocks[21], yet SOC restoration is generally slower than depletion. Land management is another crucial driver of SOC change, and several agricultural practices such as manure application[22], no-till farming[23,24], cover cropping[25] and agroforestry[26,27] have been proposed to increase SOC stocks[28]. Finally, climate change can also have a significant negative impact on global SOC stocks, by increasing SOC mineralization due to higher temperatures[29], or by decreasing carbon inputs to the soil as a result of less favorable plant growth conditions linked to more variable and extreme weather events[30]. Greater efforts in land-use and land management that turn soils into future carbon sinks are therefore required[31].

Thousands of experiments have investigated the impact of drivers of SOC change, and the findings are being consolidated in a growing number of meta-analyses[32,33]. Yet a comprehensive understanding of the global effects of land-use change, land management, and climate change on SOC is still lacking. Each of these first-order meta-analyses is restricted in its scope and often focuses on a limited number of land use interventions or geographical regions[34]. Furthermore, their results can be highly variable and sometimes contradictory, which to a certain extent is related to methodological issues and the number of experiments synthesized. A second-order meta-analysis approach combines and synthesizes results from multiple first-order meta-analyses in a systematic and quantitative way, and enhances the statistical power of the analysis by increasing sample size. It also allows for a more rigorous critical analysis, if the quality of the first-order meta-analyses is considered. Second-order meta-analysis methods have seldom been applied to drivers of SOC change (but see Bolinder et al. 2020[35], Young et al. 2021[36], and Lessmann et al. 2020[37] on the effects of some specific land-use interventions on SOC). A comprehensive synthesis of results of previous meta-analyses on SOC can facilitate evidence-based

decision-making and priority-setting aimed at increasing SOC on a global scale[38,39]. Combined with local knowledge, it can also contribute to identifying the best land-use and land management practices for SOC preservation and restoration at local and regional scales.

Here, we conducted a second-order meta-analysis that included over 220 specific drivers of SOC change, representing the direct and indirect effects of human interventions including land-use change (e.g., conversion of forest to cropland), land management (e.g., mineral fertilization), and climate change (e.g., warming). Our analysis combined 230 first-order meta-analyses (Fig. S1), that synthesized the results of more than 25,000 primary studies and 190,200 paired comparison data. In order to determine the most appropriate meta-analytical models, we examined the random structure of the models, the inclusion of quality scores, and the redundancy of primary studies (see Methods). Additionally, we compared the estimates obtained through frequentist versus Bayesian inference methods to ensure the robustness of our conclusions (Fig. S2). We found no evidence of publication bias, as indicated by the Egger's test and the trim-and-fill method (Figs. S3-5). To align with the metrics used in the retrieved first-order meta-analyses, our study examined SOC expressed as either a stock (Mg C ha$^{-1}$) or as a concentration (g C kg$^{-1}$ soil) (Fig. S6).

## Results and discussion

### Large but variable impacts of land-use change and land management, uncertain impacts of climate change

Our results revealed that the overall effects of land-use change and land management on SOC were 7–10 times larger than the direct effects of climate change (i.e., excluding the indirect effects of wildfire and snow cover change, Fig. 1). Both negative and positive effects of land-use change and land management practices were found, thereby highlighting the opportunities of increasing SOC but also the risks of its depletion.

Of the 60 types of land-use change analyzed, 25% presented a decrease in SOC that was greater than 24%, while another 25% showed an increase that was higher than 15%. Among the 143 identified land management practices, 25% displayed a decrease in SOC that was greater than 1%, while another 25% presented an increase that was

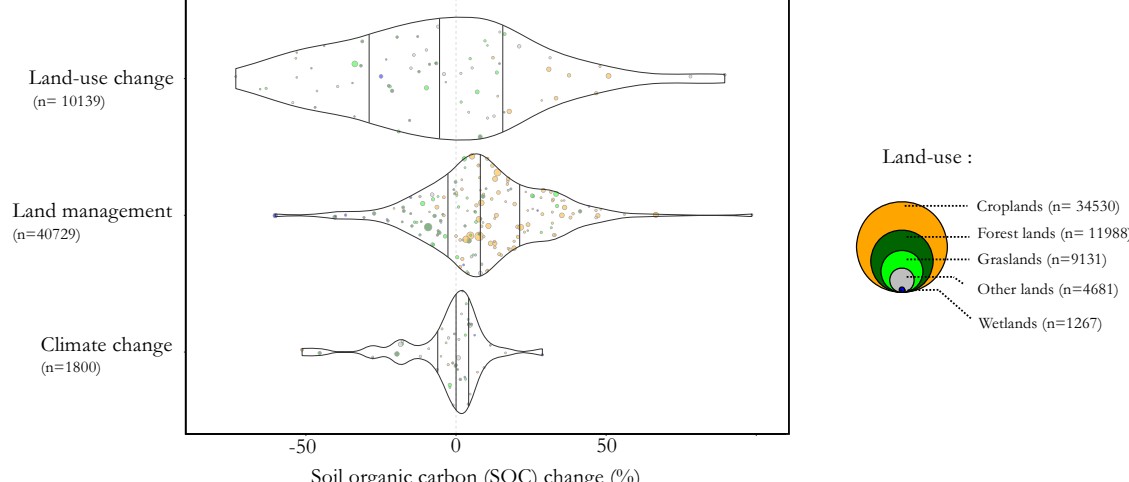

**Fig. 1 | The distribution of changes in soil organic carbon (SOC) resulting from land-use change, land management, and climate change.** Each dot represents a mean effect size in terms of SOC change for a sub-category of the three main drivers of SOC change per land-use type. Croplands are shown as orange dots, forest lands as dark green dots, grasslands as light green dots, wetlands as blue dots, and other lands as gray dots. Dots of sub-categories of land-use change are colored according to the initial (i.e., previous) land-use type. The dot sizes are proportional to the number of paired data used to calculate the mean effect sizes. Violin plots represent the distribution of values within each of the three main categories, with the 25, 50, and 75% quantiles denoted by vertical black bars. The number of paired data (n) for each land-use type is shown in the bubbles on the right side of the plot. An interactive version of the plot is available at https://rpubs.com/dbeillouin/Figure1.

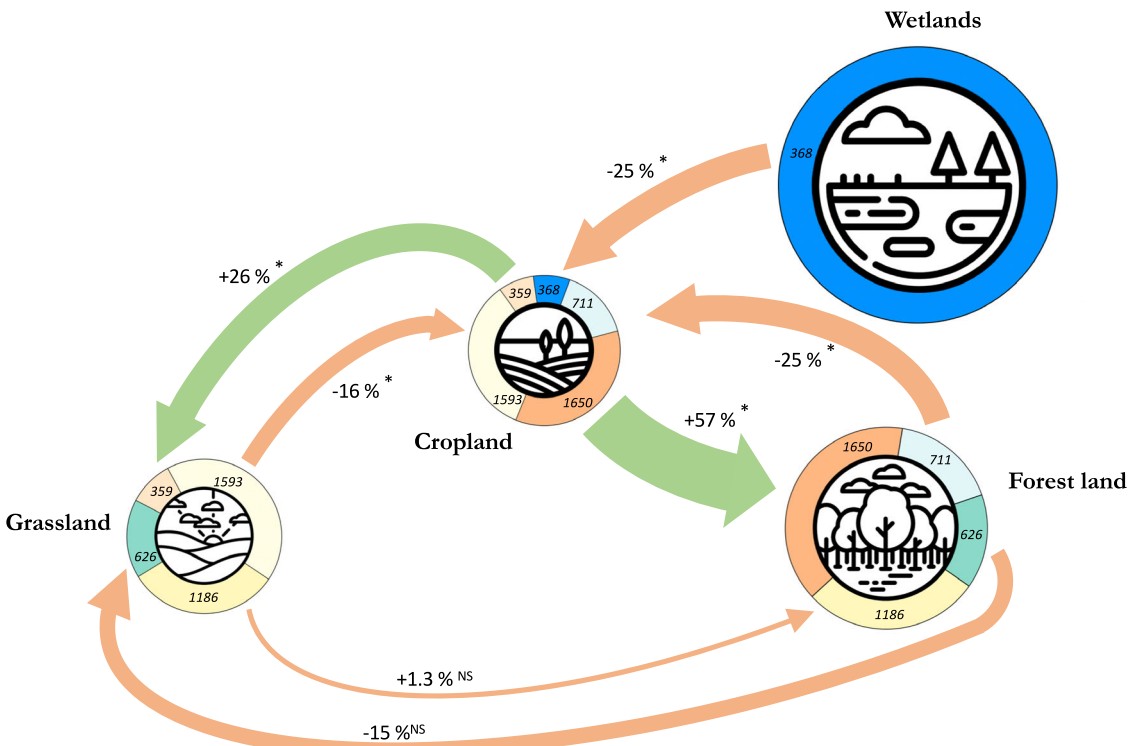

**Fig. 2 | Percentage change in soil organic carbon (SOC) due to land-use change.** The arrows represent the effect of a land-use change on SOC, with the final land-use on the arrowhead side. The arrow sizes are proportional to the magnitude of the SOC change in reference to the initial land-use, with negative effects highlighted in orange and positive effects highlighted in green. The mean SOC changes are noted alongside the arrows. An asterisk indicates a significant effect. The numbers in the disks around each land-use type represent the number of data pairs used to calculate the SOC change. The sizes of land-use pictograms are proportional to their respective mean SOC stock values (t ha$^{-1}$), as provided by FAO and ITPS (2020)[83]. Given that the mean initial SOC levels vary across different land-use types, the effect sizes (and the sizes of the arrows) expressed as percent change should be interpreted in relation to these initial levels. Details by sub-category of land-use type are available at https://rpubs.com/dbeillouin/Figure2.

higher than 23%. Thus, large but at the same time highly variable impacts can be expected, with some of the considered land-use changes or land management practices proving to be highly effective in increasing SOC, while others are not.

On the other hand, the direct effects of climate change (i.e., warming, drought, and $CO_2$ enrichment) were relatively small, as indicated by the 25$^{th}$ and 75$^{th}$ percentiles of respectively −2.4% and +4.0% SOC change. The largest climate-related SOC changes were associated with the indirect effects of climate change (e.g., wildfires, declining snow cover) (Fig. 1).

The effect of land management practices varied markedly according to the land-use type: 83% of the cropland management practices analyzed resulted in a significant increase in SOC, and 70% of all land management practices that led to a significant gain in SOC occurred in croplands (Fig. 2). On the other hand, in almost half of the cases where management practices were applied to forests, there was a significant decrease in SOC. Importantly, wetlands showed the largest decrease in SOC across all land-use types in response to management interventions (i.e., up to −60% for practices that lead to wetland degradation).

Furthermore, our results confirmed the high potential to rebuild SOC in croplands[40], which is, however, largely associated with the generally low initial SOC levels observed in croplands[17,41]. On the other hand, in the case of forest lands, the challenge is to maintain SOC levels by avoiding forest conversion and degradation[13]; few solutions currently exist to increase SOC in forest lands. Finally, the number of studies for a given land-use or land management intervention may not necessarily reflect the importance for SOC preservation. For example, there are relatively few studies on wetlands despite their crucial role as a major storehouse of carbon for climate change mitigation[16].

## The importance of carbon-rich ecosystems

Across all studies, the conversion of forest lands, grasslands, and wetlands to croplands consistently resulted in large SOC loss, with a mean change of −25%, confidence intervals (CI): [−34, −16]), −16%, CI [−30, 1.4], and −25% CI [−32, −17], respectively (Fig. 2). These large losses of SOC, combined with the extensive areas converted to croplands observed in the last decades (~ 1.0 million km² over the last 60 years)[19], have thus substantially contributed to the atmospheric $CO_2$ increase[2].

Our figures may underestimate the actual SOC losses from the conversion of forest lands because most of the underlying primary studies—and thus most of the resulting meta-analyses—quantified these losses within a time frame ranging from a few years to a maximum of a few decades following conversion, whereas the time to reach a new SOC equilibrium after a land-use change is much longer (e.g., estimated to be about 80 years for the conversion of grassland to cropland)[42]. This underlines the importance of long-term experiments and maintaining ongoing monitoring of SOC[43]. Natural ecosystems are also known to contain more stable soil carbon stores than agroecosystems[44], and their preservation is therefore crucial for near-term climate change mitigation[45].

The degree of SOC loss following land conversion to croplands varied depending on the type of cropping system established, with lower SOC losses observed after forest conversion to croplands cultivated under agroforestry practices (−12%, CI [−19, −4.9]), or with perennial crops (−7.2%, CI [−18, +4.3]), compared to those cultivated with annual crops (−32%, CI [−38, −24]). Similar results were found for the conversion of grasslands to croplands, where SOC losses were higher for croplands cultivated with only annual crops (−19%, CI [−27, −10]), compared to agroforestry or the inclusion of perennial crops (+1.7%, CI

[−5.9, +9,9] and −4.7%, CI [−9.2, +0,1], respectively). Introducing perennial crop species in croplands appears to be a promising approach for mitigating the negative effects of cropland expansion on SOC. Perennial crops are known to intercept higher amounts of solar radiation throughout the growing season and produce more biomass compared to annual crops[45], as well as allocate more resources to belowground plant parts and have permanent and deeper root systems[46]. These characteristics make them more conducive to increase SOC compared to annual crops. In addition to their benefits for SOC accumulation, perennial crops are recognized for providing a range of other ecosystem services[47]. Yet, both the extent of SOC loss following the conversion of forests or grasslands to croplands and the potential mitigation effect of perennial crops depend to a large extent on local pedoclimatic conditions[48,49].

The conversion of croplands to forests or grasslands resulted in significant SOC gains of +57% (CI [+30, +90] and +26% (CI [+8,2, +47], respectively (Fig. 2). This potential for SOC increase is particularly pronounced for degraded croplands (with low SOC levels), and generally in tropical regions[50]. However, it is widely acknowledged that ecosystem restoration often fails to fully recover the functions of the undisturbed ecosystems[51,52], including soil carbon sequestration[53]. It should be noted that a high percentage increase in SOC following the conversion of croplands to grasslands or to forests does not necessarily indicate that the SOC levels in natural grassland or forestland can be easily achieved. The high levels of SOC increase should be interpreted with caution, as the initial levels of SOC in croplands are generally low.

Finally, the conversion of forest lands to grasslands and the conversion of grasslands to forest lands did not lead to significant SOC changes (−15%, CI [−28, +1.4], and +1.3% CI [−6.4, +9.6], respectively). The effect of these two types of land-use change on SOC is considered to be particularly determined by local soil and climate conditions[54].

## Soil organic carbon increase through land management practices

Exogenous carbon inputs resulted in the largest increases in SOC in croplands and grasslands. Biochar led to a mean SOC gain of +67%, CI [+31, +112] in croplands and +32%, CI [+26, +38] in grasslands. Organic amendments applied in croplands resulted in a SOC gain of +29%, CI [+15, +45], and +34%, CI [+21, +48] in grasslands (Fig. 3, see details by sub-types of amendment in the online Table associated to Fig. 3 and in Supplementary Dataset 1). Biochar application is regarded as having a high climate change mitigation potential[55], which is supported by our findings. However, we have also demonstrated a significant variability in the effect of biochar, which can likely be attributed, in part, to the diverse application rates and physicochemical properties of biochars[52]. The application of biochar has long-lasting effects that persist for a longer period than the biomass it is derived from, resulting in most of the $CO_2$ removal benefits associated with its use. This is even true when taking into account GHG emissions that occur during its production and handling[56]. On the other hand, the scarcity of biomass in some regions or its competition with livestock feeding, particularly in sub-Sahara African countries[56], can hamper the large-scale production of biochar. Besides, possible adverse effects of biochar on soil properties and biodiversity should be considered[57]. It is also essential to emphasize that the effectiveness of exogenous carbon inputs in mitigating climate change may be restricted depending on the alternative fate of the amendments[58,59], and the feasibility to produce them. On the other hand, enhanced soil health and plant productivity resulting from organic amendments can further increase carbon inputs from plant growth and reinforce the positive effects on SOC. The use of mineral fertilizers may involve similar mechanisms. However, our results show that application of mineral fertilizer resulted in smaller effects on SOC:

+9.4%, CI [+6.3, +12.6] in croplands and +3.6%, CI [0.5, +6.7] in grasslands. Fertilizer use is considered by some authors as a main contributor to soil carbon sequestration (an estimated global increase of 70–88 Mt C yr$^{-1}$, Lessmann et al. 2022[35]), especially in low fertile soils[60]. Interestingly, our study suggests that partial or total substitution of mineral fertilization with organic amendments in cropland leads to an increase in SOC of +34%, CI [+20, +49].

Agroforestry significantly increased SOC in croplands by +20%, CI [+17, +23]. The integration of trees in croplands resulted in an average SOC increase of 33%, CI [+24, +43] for multi-strata systems, 32%, CI [+9.0, +60] for parklands, and 21%, CI [+14, +30] for alley cropping, 19%, CI [−4.9 + 50] for improved fallows, and 17%, CI [+13, +22] for hedgerows (see the online interactive Table associated to Fig. 3). Growing trees in grasslands (i.e., silvopasture) also resulted in a significant SOC increase of +26%, CI [+11, +42]). The use of agroforestry is regularly brought up by policymakers as a climate change mitigation action, e.g., 40% of the 147 non-Annex I countries under the Kyoto Protocol propose agroforestry as a solution in their Nationally Determined Contributions[61]. Yet much of the SOC storage potential associated with the integration of trees in croplands and grasslands occur in countries where agroforestry is not considered as a climate change mitigation option[62]. Moreover, including trees in croplands could increase competition for resources with crops and often requires supplementary nutrient inputs in the short term[63]. It is worth noting that other types of crop diversification had limited effects on SOC in croplands. For instance, crop rotation resulted, on average, in a 6.5% SOC increase (CI [−0.9, +14]), and mixtures of crop species showed a 0.2% change (CI [0.1, +0.3]), on average. Interestingly, replacing a monospecific forest with a mixture of tree species resulted in a significant 9.0% SOC increase (CI [+5.4, +12.7]). The number of meta-analyses investigating the effects of some of these management practices on SOC is, however, limited.

Several other land management practices showed smaller but significant positive changes in SOC, including no-till farming (+9,3%, CI [+5,6, +13]; see the online interactive Table associated to Fig. 3), reduced tillage intensity (+12%, CI [+0.1, +24], crop residue retention (+13%, CI [+9.8, +16]), and perennial energy cropping (+12, CI [+6.8, +17]). Several of the above practices are frequently implemented in a combined form. An example of combining several practices is the use of reduced tillage, crop residue retention and crop diversification, which are the fundamental principles of conservation agriculture. However, the number of first-order meta-analyses dealing with combined and more complex agricultural practices remains limited (Supplementary Dataset 3). Similarly, organic agriculture involves varying degrees of organic amendments and crop diversification, among other practices. Our results indicate a large effect of this farming system on SOC (+35%, CI [+11, +64]).

In contrast to the aforementioned practices, some other land management practices had significant negative effects on SOC, such as prescribed fire in forest lands (−21%, CI [−34, −4.9]). It is worth noting that prescribed burning is sometimes recommended to reduce carbon losses from possible future wildfires, but its carbon costs (i.e., GHG emissions and SOC losses) may outweigh the reduced wildfire emissions[64]. Some other forest management practices (i.e., converting secondary forest to plantation; −23%, CI [−24, −21%]), forest harvesting (−8.1%, CI [−13, −3.1]) also negatively impacted SOC. In grasslands, increased intensity of grazing (−9.9%, CI [−18, −0.5]) and the presence of grazing compared to no grazing (−7.1%, CI [−31, −3.9]) had negative impacts on SOC.

## Indirect effects of climate change could have a pronounced negative effect on SOC

Direct effects of climate change such as warming, drought or rainfall increase, and $CO_2$ enrichment, had either uncertain or small impacts on SOC change when analyzed individually (Fig. 4). Specifically,

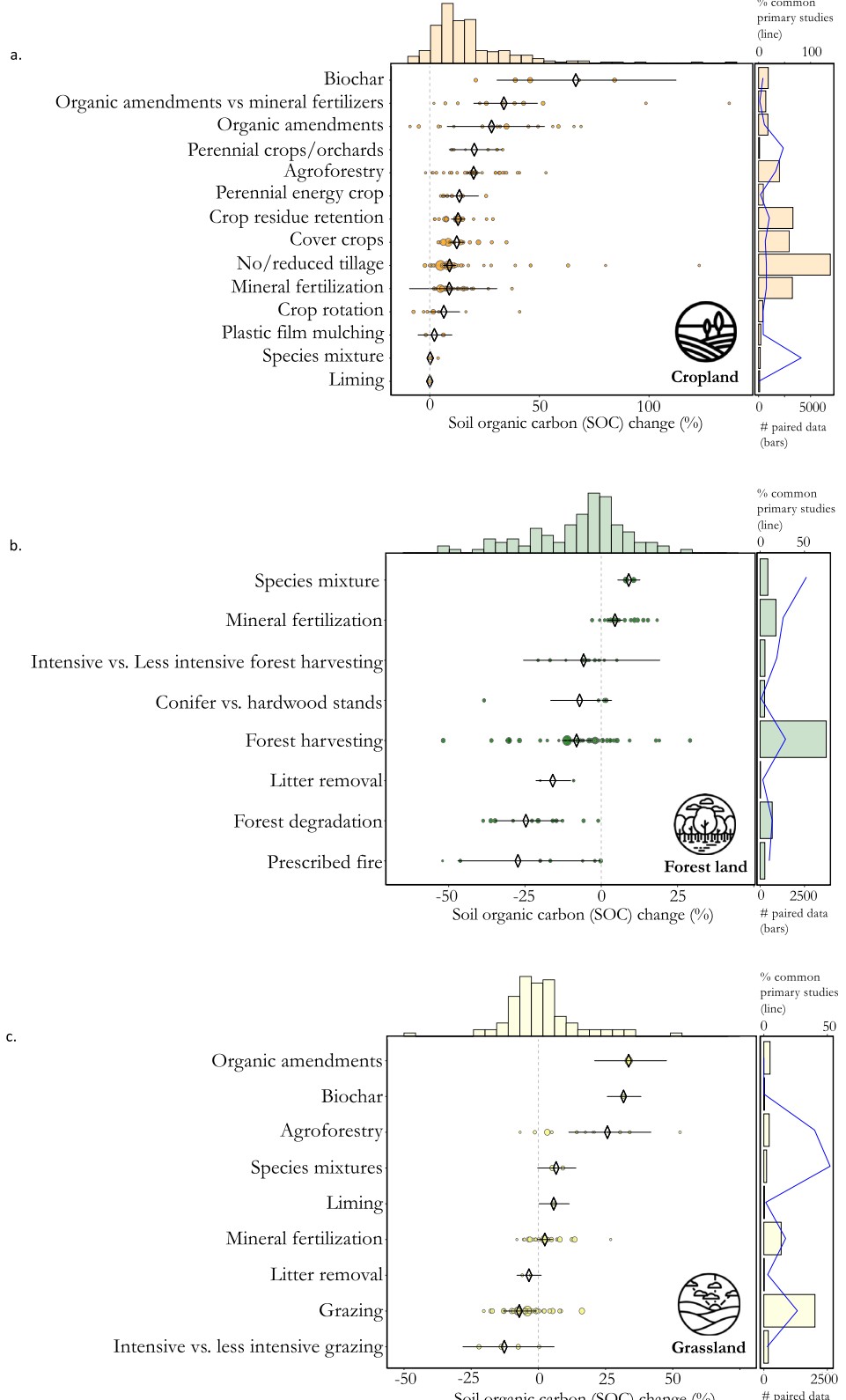

**Fig. 3 | Percentage change in soil organic carbon (SOC) due to land management practices.** Results are detailed for for croplands (**a**) forest lands (**b**) and grasslands (**c**) Diamonds and lines of the main plots represent the mean effect sizes and 95% confidence intervals (CIs), respectively. The dot sizes are proportional to the number of paired data analyzed. The histograms above each plot represent the probability distribution of effects for all practices combined. The bar plots on the right side of each main plot represent the number of primary studies for each practice (bars) and the percentage of primary studies used by at least two meta-analyses (lines). Forest degradation includes the transformation of primary forest to secondary forest and secondary forest to plantation forest. Details of the information used to make this graph as well as details by sub-category of land management are available at https://rpubs.com/dbeillouin/Figure3.

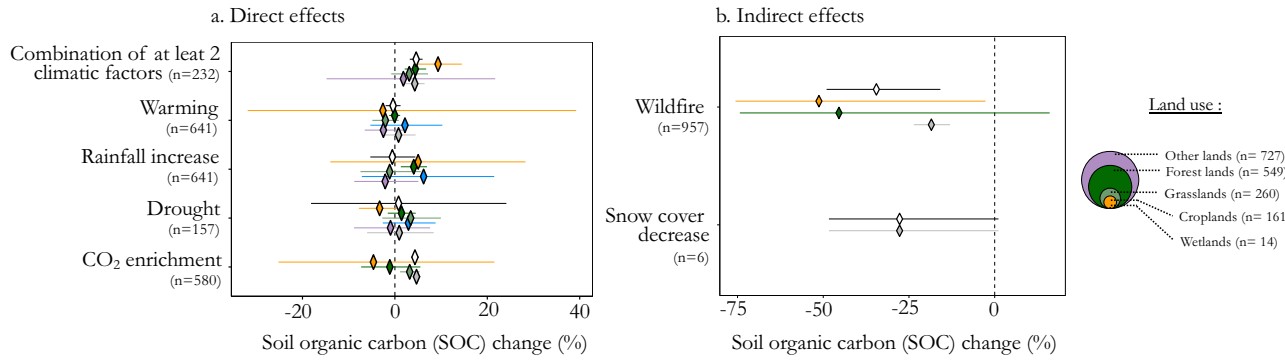

**Fig. 4 | Percentage change in soil organic carbon (SOC) due climate change.** Results are detailed for direct (**a**) and indirect factors (**b**) of climate change. Diamonds and lines represent the estimated mean effect sizes and the 95% confidence intervals (CIs), respectively. The number of paired data (n) for each category of land-use type is shown in the bubbles on the right side of the plot. Details of the information used to make this graph as well as details by sub-category of climate change factor are available at https://rpubs.com/dbeillouin/Figure4.

drought did not significantly affect SOC (mean effect across all land-use types: +0.8%, CI [−1.9, +3.7]), the same was observed for warming (−0.3%, CI [−2.3, +1.8]), and rainfall increase (+1.4%, CI [−2.3, +5.3]). $CO_2$ enrichment, on the contrary, had a significant effect on SOC for two of the four land-use types analyzed, and a mean effect of +4.3%, CI [ + 3.5, +5.2], across all land-use types.

Considering the combined direct effects of climate change, there was a significant positive impact on SOC change, with a mean effect of + 4.6%, [+3.2, +6.0]. This effect can likely be associated with the presence of $CO_2$ enrichment in the majority of the examined combinations. Available data on the individual and combined direct effects of climate change on SOC are, however, very limited. Further studies are needed to better understand the individual and combined effects of climate change on SOC[65], especially in the different land-use types and under different climatic conditions. This would allow to refine our understanding of the mechanisms controlling SOC dynamics and develop more accurate projections of SOC changes in response to global climate change. In particular, there is a need to better understand the responses of the different SOC fractions to climate change effects[42] (and see Supplementary Dataset 2), including the various confounding factors affecting SOC decomposition rates[66]. For example, climate change simultaneously affects SOC input through its effect on both plant biomass production and SOC decomposition, i.e., warming can lead to increased plant biomass, but it can also enhance decomposition;[14] elevated $CO_2$ levels can stimulate plant growth while it is expected that they have a minimal effect on SOC decomposition[67].

On the other hand, our findings indicate that indirect effects associated with climate change, such as wildfires or a decrease in snow cover, may have a more substantial impact on SOC than the direct climate change effects. The underlying data are, however, scarce and the available results should be confirmed in further experimental studies. It is worth noting that the effect of wildfire on SOC is in line with the effect observed for prescribed fires in forests (as shown in Fig. 3), as well as with the effect of fires of 'non-classified land types' (as detailed in the online Table associated to Fig. 4). These indirect effects associated with climate change should not be overlooked, given that, for instance, the occurrence of forest fires has doubled in the last 40 years[68], and has notably increased in many biomes[69]. Other indirect effects of climate change, such as the effect of flooding and changes in snow cover, have thus far received limited attention in existing meta-analyses, and there is a lack of synthesized results on these subjects.

In summary, our comprehensive second-order meta-analysis, encompassing over 220 types of land-use change, land management practices, and climate change factors impacting SOC, has identified both the main factors associated with SOC loss and various options that have the potential to maintain or increase SOC levels on a global scale. The preservation of SOC can be effectively achieved through the protection of natural ecosystems and the introduction of perennial crops in croplands. However, it is important to note that worldwide efforts to preserve SOC may encounter local challenges arising from the indirect effects of climate change, which may have a more substantial impact on SOC compared to direct climate change effects.

In our analysis, we have identified significant knowledge gaps that require further research, namely the neglected ecosystems such as wetlands and the impact of climate change on SOC. Our global second-order meta-analysis is a valuable tool for scientists conducting future research on SOC preservation and restoration. The results of our study, including the ranking of effective land management practices for protecting or increasing SOC, can serve as a valuable reference for decision-makers involved in climate change mitigation efforts. However, it is essential to exercise caution when implementing these results in local contexts. This is due to the unique pedo-climatic conditions of different regions and the limitations in the representativeness of our results in certain environments.

## Methods

We conducted a second-order meta-analysis to examine the impact of drivers of SOC change, namely land-use change, land management and climate change, across various land-use types. Our analysis synthesized the findings of 230 first-order meta-analyses on SOC that were conducted worldwide (see Fig. S1 for details) on croplands, forest lands, grasslands, wetlands, and other lands. We assessed the quality of the first-order meta-analyses and the potential overlap of primary studies between the meta-analyses to produce reliable estimates of effects.

### Systematic literature search

We first performed a systematic search for peer-reviewed meta-analyses on bulk SOC stocks or concentrations using various databases, including the Web of Science Core Collection, Scopus, CAB Abstracts, and Agricola (through the OVID platform), and the Google Scholar search engine. The search was done on January 9th, 2020, and updated on July 10th, 2022, using the following search string: ("meta*analysis" OR "systematic review") AND ("soil organic carbon" OR "SOC" OR "soil organic matter" OR "SOM" OR "soil carbon") in title, abstract, and keywords fields. We screened the titles and abstracts of the 1005 identified papers for their potential inclusion in our study. The full text of the retained papers was then independently reviewed by at least two co-authors of this study for the following criteria. A paper had to (i) analyze the effect of one or several factors on bulk SOC stocks or concentrations, (ii) present a statistical formal analysis of at least two primary studies on SOC, (iii) present indicators of precision of the

effect sizes (standard errors or confidence intervals). At the end, 230 meta-analyses were retained for our second-order meta-analysis. Further details on the methods used can be found in a related data paper[34] and in an evidence map[33].

## Data extraction and coding

All the effect sizes of the 230 meta-analyses included in our study were then extracted from the text, tables, or figures (using Plot Digitizer http://plotdigitizer.sourceforge.net/) and recorded in an Excel file. The land-use type associated with each effect size was documented according to IPCC standards[70]. We also extracted the metrics (e.g., ratio, percentage change), possible transformations (e.g., logarithm), confidence intervals or other indicators of variability, and the number of primary studies and observations that were used to calculate the effect sizes. The list of primary studies (and their DOIs, when available) used in each meta-analysis was retrieved, thereby allowing us to identify the number of common primary studies between each pair of meta-analyses. Finally, we characterized the meta-analyses included in our study by eight quality criteria related to the literature search, statistical analyses, and potential bias analysis (see Beillouin et al., 2021[34] for a precise description of these criteria). A poor methodology used to retrieve the primary studies and analyze the data can indeed result in biased and misleading results[71].

Effect sizes (and confidence intervals) expressed as standardized mean differences or percentage changes in the first-order meta-analyses were converted into ratios to ensure the comparability of the results between the meta-analyses[72] and were log-transformed to ensure normality. After applying the meta-analytical models (see below), we transformed the log ratios and their associated confidence intervals and expressed them in percent change to facilitate interpretation. We categorized the drivers of SOC change reported in the meta-analyses into three main categories: land-use change, land management, and climate change. In case a meta-analysis presented subgroup analyses, only independent effect sizes, i.e., based on different sets of primary studies, were retained.

## Pairwise meta-analysis

To estimate the effect of factors on SOC, we tested several meta-analytical models, varying in the structure of their random effects, the inclusion or not of the quality score of the first-order meta-analyses, and of the redundancy of the primary studies between the meta-analyses. The best model for each factor (whose results are presented in the main text) was then chosen based on the Akaike information criterion (AIC).

The most complex model is a three-level meta-analytical model, including a variance–covariance matrix considering the precision, the quality and the redundancy of the first-order meta-analyses. This model is written as follows:

$$\log(Y_{ij}) = \mu + b_i + \varphi_{ij} + \varepsilon_{ij} \tag{1}$$

with $b_i \sim N(0, \tau^2)$, $\varphi_{ij} \sim N(0, \upsilon^2)$ and $\varepsilon_{ij} \sim N\left(0, \sigma_{ij}^2\right)$, where $Y_{ij}$ is the $j^{th}$ effect-size of the $i^{th}$ meta-analysis (one meta-analysis could present several effect-sizes), $\mu$ is the mean estimated effect (shown as the diamonds in the Figures), $b_i$ is the random meta-analysis effect (i.e., between-cluster heterogeneity), $\varphi_{ij}$ is the random effect size effect within the $i^{th}$ meta-analysis (i.e., within-cluster heterogeneity), and $\varepsilon_{ij}$ is the random estimation error associated with the $j^{th}$ effect size of the $i^{th}$ meta-analysis (i.e., the sampling error). Here, the clusters represent the meta-analyses included in our study. The three-level model implies the estimation of two heterogeneity variance parameters (here noted $\tau^2$ and $\upsilon^2$).

We weighted each effect size by the inverse of its variance, as recommended by Marín-Martínez and Sánchez-Meca (2010), and reduced the weight of the lower-quality meta-analyses according to

Doi et al. (2015)[73,74]. As a quality proxy, we used the percentage of the eight quality criteria met by a meta-analysis (see Beillouin et al., 2022[33] for a detailed explanation of the criteria). We also considered the non-independence between the effect sizes of different meta-analyses by calculating a variance-covariance matrix based on a pseudo correlation between meta-analyses[75]. The proxy of the correlation between each pair of meta-analyses was estimated as $(2 \times m)/(n_1 + n_2)$, where m is the number of common primary studies between each pair of meta-analyses, and $n_1$ and $n_2$ represent the total number of primary studies in the two respective meta-analyses.

The other meta-analytic models that were tested corresponded to (i) a model with a simplified random structure, i.e., without the between-cluster heterogeneity ($b_i$); (ii) a three-level hierarchical model without considering the redundancy of primary studies, and (iii) a three-level hierarchical model without considering both the redundancy of primary studies and the quality of the first-order meta-analyses.

## Analysis of the results and sensitivity analyses

The model results were summarized by the median as a point estimate and the 95% confidence interval as a measure of uncertainty of the point estimate. Potential publication bias was assessed with funnel plots and Egger's test[76]. The funnel plots assume that studies with high precision will be plotted near the average mean effect, and studies with low precision will be spread evenly on both sides. Egger's test gives the degree of funnel plot asymmetry as measured by the intercept from regression of standard normal deviates against precision. The sensitivity of the results against publication bias was tested with the Rosenthal fail-safe number, i.e., the number of additional studies with a mean null result necessary to provide a non-significant global estimated effect (see Fig. S3-5). We also estimated the mean effect considering the missing studies based on the trim-and-fill methodology[77].

Finally, we tested the robustness of our model results by comparing the results obtained using frequentist and Bayesian statistics. Frequentist model parameters were estimated by maximum likelihood using the *metafor* R package[78]. For the Bayesian approach, we used weakly informative prior scenarios, following the recommendations of Williams, Rast, and Bürkner (2018)[79]. Specifically, the true pooled effect size prior was set to a normal distribution with a mean of 0 and a variance of 1. The variance priors of the models were set to a half-Cauchy distribution with location parameter set at 0 and a scale parameter set at 0.5. The posterior distribution was approximated through Markov chain Monte Carlo (MCMC) simulation methods using Just Another Gibbs Sampler (JAGS) software (version 4.3.0) through the *brms* package[80]. The MCMC algorithm was run with three Markov chains, each including 20,000 iterations after a burn-in period of 8,000 iterations. In addition, the chains were thinned by storing one out of ten iterations to reduce autocorrelations in the subsequent sample. Convergence was assessed via three different criteria: (i) the potential scale reduction factor, $\hat{R}$, whose values must be equal or close to 1, (ii) the effective number of independent simulations draws, *neff*, which must be >100, (iii) graphically, by drawing trace plots and assessing whether the simulated values of the chains overlapped. The best Bayesian model for each factor was chosen based on the wAIC[81]. Comparisons of the model results between the frequentist and Bayesian approach are available in Fig. S 2.

All statistical analyses were conducted with R software (version 3.0.2), dplyr for data management, and ggplot2 for data visualization. All scripts used in this study are available in the MetaSynthesis R[82].

## Data availability

The data are available under the repository https://doi.org/10.18167/DVN1/KKPLR8 and described in the corresponding DataPaper: Beillouin et al. 2021[34].

## Code availability

The codes used for the analyses are available in the MetaSynthesis package[82]

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

## Acknowledgements

Support for this study to M.C. was provided through the CGIAR initiative Mitigate + : Research for Low-Emission Food Systems, and through the contributions of all funders to the CGIAR Trust Fund.

## Author contributions

D.B.: Data retrieval, formal analysis, draft, review. M.C.: Data retrieval, draft, review. J.D.: Data retrieval, draft, review. DavidB: Data retrieval, review. A.B.: Data retrieval. A.F.: Data retrieval, review. F.F.: Data retrieval, review. R.C.: Data retrieval, draft, review.

## Competing interests

The authors declare no competing interests.
