## [Peer Review File · Nature Communications]

A Global Meta-analysis of Soil Organic Carbon in the AnthropoceneReviewer #1 (Remarks to the Author):

In this study effects of anthropogenic activities such as land use changes, land management and climate change on SOC were derived on the basis of a meta-analysis of 184 first-order meta-analyses covering more than 12,000 primary studies. Although the number of underlying data is impressive and a second-order meta-analysis of numerous first-order meta-analysis may generally increase the statistical power, I cannot see really novel findings in this manuscript. For example, that a conversion of extensive land uses such as grassland or forest to cropland leads to a SOC loss or that SOC can be increased by improved land management practices such as agroforestry systems or addition of external carbon sources are well-established facts. From my perspective, a second-order meta-analysis would be interesting if any new aspects emerge, but just a confirmation of well-known facts does not justify publication, even when based on a large data set. Therefore, I recommend to reject this manuscript.

Reviewer #2 (Remarks to the Author):

The research work reported in this manuscript was aimed at examining the effects of land use changes, land management practices and climate change on soil organic carbon (SOC). The authors have undertaken a comprehensive meta-analysis to quantify the effects of various land management practices and climate change-induced extreme weather events (drought/warming etc) on changes in soil carbon under various land use systems that include cropping, pasture, wetlands and forestry. Based 184 meta-analysis data, they have demonstrated that: (i) the conversion of any land use to croplands results in high SOC loss (ii) most land management practices affecting established forests deplete SOC, and (iii) indirect effects of climate change, such as wildfires, appear to have a greater impact on SOC than direct effects (e.g. rising temperatures). The outcomes from this research work are closely aligned the scope of the Journal of Nature Communications, and hence the manuscript is suitable for publication in this journal. However, the authors need to clarify following points before it can be accepted for publication.

- **Line 54: May like to rephrase 'Soil organic matter (SOM) which comprised mainly of carbon is a key component of soils' (soil quality and soil health are regulated by SOM – not carbon perse).**
- **Line 60: May like to rephrase 'For this reason, decarbonizing soil has recently been ...mitigation. (<https://www.fao.org/3/ca6522en/CA6522EN.pdf>)**
- **Line 72 (may like to include this reference – (Ramesh, T., Bolan, N. S., Kirkham, M. B., Wijesekara, H., Kanchikerimath, M., Srinivasa Rao, C., . . . Freeman, O. W. (2019). Soil organic carbon dynamics: Impact of land use changes and management practices: A review. In D. L. Sparks (Ed.), *Advances in Agronomy* (Vol. 156, pp. 1-107). Cambridge, MA: Elsevier. doi:10.1016/bs.agron.2019.02.001)**
- **Line 78-81. Crop residues and no-till farming are important land management practices which impact SOM (Chowdhury, S., Farrell, M., Butler, G., & Bolan, N. (2015). Assessing the effect of crop residue removal on soil organic carbon storage and microbial activity in a no-till cropping system. *Soil Use and Management*, 31(4), 450-460. doi:10.1111/sum.12215 Mehra, P., Baker, J., Sojka, R. E., Bolan, N., Desbiolles, J., Kirkham, M. B., . . . Gupta, R. (2018). A Review of Tillage Practices and Their Potential to Impact the Soil Carbon Dynamics. In D. L. Sparks (Ed.), *Advances in Agronomy* (Vol. 150, pp. 185-230). Cambridge, MA: Elsevier. doi:10.1016/bs.agron.2018.03.002)**
- **Line 107: Carbon Stock (Mg C/ha) – is it based on soil mass or soil volume**
- **Line 131: There are two types of forests – i.e., managed (agroforestry/planation forest) and unmanaged forests (parkland/native); perhaps may like to make a statement on this; forest land management practices apply only to managed forests**
- **Line 138: Most wetlands are unmanaged; therefore, land management practices may not apply for wetlands – which is reflected in Figure 2 – no change in wetland land use (the last column in that figure)**
- **Line 203: In relation to degraded land, one important land use change that has not**

been covered in this research work is mine-site revegetation which is increasingly used for carbon sequestration and carbon storage (Wijesekara, H., Bolan, N. S., Vithanage, M., Xu, Y., Mandal, S., Brown, S. L., . . . Surapaneni, A. (2016). Utilization of biowaste for mine spoil rehabilitation. In *Advances in Agronomy* (Vol. 138, pp. 292 pages). doi:10.1016/bs.agron.2016.03.001). Perhaps the authors may like to include a statement on the value of mine-site for carbon storage

- Line 245: Fertilizer input is a major contributor to soil carbon sequestration, especially in low fertile soils (Chowdhury, S., Bolan, N., Farrell, M., Sarkar, B., Sarker, J. R., Kirkham, M. B., . . . Kim, G. H. (2021). Role of cultural and nutrient management practices in carbon sequestration in agricultural soil. In *Advances in Agronomy* (Vol. 166, pp. 131-196). doi:10.1016/bs.agron.2020.10.001)

- Line 282: liming is an important land management practice to enhance soil carbon sequestration in acid soils (Pedro Luiz Oliveira de Almeida Machado¹, Vinicius de Melo Benites, Nanthi Bolan. 2021. 21. Liming Acidic Soils. RECARBONIZING GLOBAL SOILS A technical manual of recommended management practices. VOLUME 3: CROPLAND, GRASSLAND, INTEGRATED SYSTEMS AND FARMING APPROACHES PRACTICES OVERVIEW. Food and Agriculture Organization of the United Nations Rome, 2021, 289-303)

- Figure 4: Climate change can lead to increasing intensity and frequency of extreme weather events including drought, flood and bushfire. The authors have covered the effects of drought and bushfire on soil carbon; flooding can cause a decline in Soil carbon through leaching of dissolved organic carbon and erosion of sediments; perhaps the authors may like to include a statement on that.

Reviewer #3 (Remarks to the Author):

Overall, an impressive study with highly relevant results. The authors used a second order meta-analysis and nicely show the distinct effect of either converting one land-uses to another or applying an improved management technique on an established land use. I think the strength of this study is to give global guidance on which conversions work and which managements are best practice. I see two caveats that result from the structure of the study, which I think should be highlighted in the discussion of results to avoid misinterpretation by the audience 1) through assessing relative change of SOC from one land use to the other, the results only apply when one land use is replaced by another and the basis of SOC is always the current level. It should, for example, be avoided that people get the impression that more carbon can be gained from afforestation of cropland than is lost from deforestation for cropland (Figure 2 could give that impression, if not looked at carefully). 2) The joint analysis of a global dataset is at the same time strength and weakness of the study. It gives a very nice overview on what global policy should focus on and at the same time masks small scale heterogeneity, which is still highly relevant for local implementation. The discussion would benefit if this issue would be addressed and if the authors would delineate more clearly for which purpose the study serves best (IMHO setting global priorities), and what it cannot tell with as much certainty (how it will play out at the local scale). I do think that the manuscript does need some medium revision, e.g. some parts of the methodology (exact inclusion/exclusion criteria, statistical model) are not expressed with sufficient detail, and I also found some numbers that are probably wrong in the text. See my specific comments for details. However, after successfully addressing those comments, I think the study will be of great value and deserves to be published in nature communications. A final suggestion: The authors may think of making the data and R code of this article available upon publication. It is a highly interesting dataset to the scientific community and would likely increase the visibility of the article!

Specific comments:

L65 – particularly essential in which way?

L67 – Not sure what is the link between irrecoverable carbon emissions and negative

emissions. Is 165-67 essential to your work or could you delete this?

L70 -71 The percentages here somehow contradict with your introductory sentences that SOC is the most important terrestrial C storage

L77 – Would be good to mention the asymmetry here. SOC is lost much faster than it is rebuilt.

L79 – manure application

L81 – You mean that greater efforts may be needed in the future?

L89 – it is not fully clear to me what is meant with this sentence. If some NET are country specific, can such higher order analyses not mask this? I agree that such higher order analyses are valuable to prioritize. At the same time, local implementation should be done on local best knowledge.

L94 – while I agree that the higher sampling size increases statistical power, I think that flattening out small scale variability (part of it true variability, part of is measurement error) does not only have benefits. You should also acknowledge that to some part this eliminates existing small scale variability. A powerful example how this can also lead to the wrong conclusion is given in Bradford et. al. (2021)

<https://link.springer.com/10.1007/s10533-021-00789-5>

L98 – Please give a definition of what is a driver for you.

L104/105 – This sentence is not clear to me. I think you should give a very brief but exact description here of what model you use. I do not understand the sentence about frequentist of Bayesian, usually it must be one of these. Which one did you use?

L107 – Good that you included both, but how did you deal with the difference? Transfer one into the other? Did you consider that there is a difference in data quality i.e. stocks based on equivalent soil mass > SOC stocks > SOC contents?

L111 – Could you make this title more specific?

L114-126 – I think it is good that you start with an overview of your data. At the same time I think this paragraph should be shortened, as the only really interesting thing is that climate change impacts are smaller than LUC and MGMT. That the latter two have a large range as there is e.g. both afforestation and deforestation included is not surprising and does not require that much text, in my opinion.

L127 – A definition of “main land management practices” should be given.

L137- Can you refine what you mean by that? That deforestation has a strong loss of SOC?

Figure 2 – Is there an error with the confidence intervals of grassland > forest? Median is -3%, while CI is +10 to +4%.

L172- Just significant SOC loss?

L181/2 – Anthropized?

193- interesting option instead of means?

L207/8 - This very important point should be highlighted even more, maybe with a numeric example with absolute.

L210 -) missing

Figure 3 – Could you rename energy crop, else it sounds like it includes biogas maize (which I guess it is not biogass maize)

L226 – two dots

L237 – you should also address that organic amendments are often already used. Hence higher amendments at scale are often not even possible as they are limited by NPP.

Biochar, as much more stable may be the exception here.

L239/40 – Not sure if this second order effect really adds to the discussion here.

L243/44 – Were you not talking about the effect of replacing mineral before? So why these two sentences here?

L250-52 – I disagree here. NPP is high in many areas of SSA. So there should be enough feedstock for biochar. Additionally, it has been proven highly effective in improving soil fertility (Kätterer, et al; 2022; <https://doi.org/10.1007/s13593-022-00793-5>)

L265 – This could be discussed more. Combined with the results of manure addition, I would interpret it as the amount of C input being the most relevant (less than diversity).

L266 – highly significant meaning $p < 0.001$? This is the only time you use this term.

L274 – I suggest you add a “Yet,” before “The effect of mixed species . . .”

L278 – How is no-till farming significant if the CI is from -1.6 to 2? The 9.2% cannot be, some of the numbers must be wrong.

L309 – For grassland, CO2 enrichment actually had a significant positive effect, right?

L310/11 – Maybe for that reason you should write in this section that you did not find significant effects instead of that there “were” none.

L313 – Contrasting?

L318 – This makes a lot of sense! Very interesting.

L330 – I think it is worth to mention how they should best be interpreted. In my opinion they serve as an important overview on what works best. Yet at smaller scales, it may still be advisable to look for local studies when planning land use, as for example indicated by your outliers of Fig. 3.

L373&380 – Duplication is an important point. Did you somehow check, whether the meta-analyses you used did not use the same studies. I assume if there is a huge overlap between two meta-analyses one should be excluded. Else the same data is ruminated many times. You have not specified how you handled overlap.

L381 – would be good to also state these quality criteria

L386 – I am confused by your statement that you used ratios, as your outputs are given in %change from the baseline.

L399/400 – I do not fully get this hierarchical model. It would be good to describe this better with some numerical examples. What is the difference between φ and ε ? From your description, it seems that they are the same, e.g. you have a random effect for each effect in each meta-analyses, but how can you distinguish it from the residual error term if they have the same number of observations. At least from your description it seems like that is the case.

L412 – I miss information on which software and statistical packages were used to implement the analysis. Also, I am still confused about the frequentist vs Bayesian statement from the introduction? What method did you finally use? I now saw it is written below, but still have the feeling it should come earlier.

L415 – As you worked with transformed data – how did you backtransform the confidence intervals? As they are uneven, it looks like you did it correctly, but it does not hurt to state the details here.

LI425ff – It is good that you tested the robustness and it would be even better if you share with the reader if your models were in fact robust. Same goes for the Bayesian vs frequentist, below.

We thank the three reviewers and the associate editor for their critical and constructive review of our work and the overall positive comments. We have meticulously reviewed the text and our analysis in response to the criticisms and comments. In particular, with our revised version, we have :

1. Updated the database to include newly published meta-analyses on the subject. Meta-analyses from 2020 to mid-2022 have now been incorporated, resulting in a total of 230 retrieved studies (instead of 184 previously). Our aim was to provide the most up-to-date information possible. This update confirms our results and allows us to be more precise in our analyses.
2. Created an R package to make the analyses fully transparent and reproducible. The package is available at : <https://github.com/dbeillouin/MetaSynthesis>. We have also clarified our methodology for selecting the best models for each analysis as requested by the reviewer 3.
3. Improved the graphical presentations to better convey our results. We believe that new graphic representations can reach a wider audience while being clearer. In addition, we have complemented each figure with an interactive table or figure that is available online. This allows full transparency of our results and allows researchers to dig into the details that we could not display in the main text
4. Improved the text so that the final result is clearer and more readable manuscript, also by incorporating the reviewers' comments. We have, for example, better clarified the purpose and use of our work (i.e. support to public decision making, cf reviewer 3), and also presented more clearly the knowledge gaps to help the scientific community to advance more directly in this field. we have also slightly modified the title of our article which is now '**Soil Organic Carbon in the Anthropocene: A Global Meta-analysis**'

A detailed point-by-point response to the comments can be found below:

REVIEWER COMMENTS

Reviewer #1 (Remarks to the Author):

In this study effects of anthropogenic activities such as land use changes, land management and climate change on SOC were derived on the basis of a meta-analysis of 184 first-order meta-analyses covering more than 12,000 primary studies. Although the number of underlying data is impressive and a second-order meta-analysis of numerous first-order meta-analysis may generally increase the statistical power, I cannot see really novel findings in this manuscript. For example, that a conversion of extensive land uses such as grassland or forest to cropland leads to a SOC loss or that SOC can be increased by improved land management

practices such as agroforestry systems or addition of external carbon sources are well-established facts. From my perspective, a second-order meta-analysis would be interesting if any new aspects emerge, but just a confirmation of well-known facts does not justify publication, even when based on a large data set. Therefore, I recommend to reject this manuscript.

We thank reviewer 1 for his analysis and critical review of our paper. This is indeed a common and logical criticism of first-order or second-order meta-analyses. However, we aimed at summarizing scientific knowledge by reusing research data in a transparent way, and we believe that our work makes a significant contribution to the field of research. In more detail, we believe that:

- To reuse the example used by reviewer 1, even if it is well known that the conversion of forests or grasslands to croplands leads to SOC loss and that on the opposite agroforestry or organic amendments can store SOC, the percent change is quite uncertain. With this large, up-to-date compilation of data, we believe that we provide robust estimates that were not achieved so far.
- In addition to compiling a comprehensive list of published meta-analyses on soil carbon, our work also presents a database and interactive tables for exploring the effects of different interventions, subtypes of interventions, and their moderators. This level of transparency and accessibility is crucial for promoting the re-use of research results and can serve as a benchmark for future studies on soil carbon.
- Furthermore, our work is not just a simple compilation of studies, but a rigorous statistical analysis that takes into account the precision and quality of each study. Methodological differences can impact the results. Therefore, our analysis provides an highly accurate estimation of the average effects of each intervention, which is essential for making informed decisions to improve the stock of soil carbon. Our results provide estimates that were not available in the literature and are difficult to estimate even by experts in the field; it is impossible to objectively interpret several meta-analyses and the effects of dozens of agricultural practices by experts.
- Importantly, our study does not only estimate the average effect of one or two interventions but of all the interventions so far analyzed in the scientific literature. This allows readers to gain a broader understanding of the most effective practices for increasing soil carbon, and rapidly identify the most promising techniques to increase soil organic carbon globally (or identify main threats). We also highlight gaps in knowledge and thus provide a rationale for future research and research project development.

To respond to the criticism of reviewer 1, we think that our study provides timely and important findings for the scientific community and those interested in soil carbon storage. For example, we found that the quantitative direct effects of climate change on soil carbon are, on average, weaker than the effects of land use change and agricultural practices, and indirect effects of climate change. These findings have significant implications for climate change mitigation efforts and soil management strategies.

Reviewer #2 (Remarks to the Author):

The research work reported in this manuscript was aimed at examining the effects of land use changes, land management practices and climate change on soil organic carbon (SOC). The authors have undertaken a comprehensive meta-analysis to quantify the effects of various land management practices and climate change-induced extreme weather events (drought/warming etc) on changes in soil carbon under various land use systems that include cropping, pasture, wetlands and forestry. Based 184 meta-analysis data, they have demonstrated that: (i) the conversion of any land use to croplands results in high SOC loss (ii) most land management practices affecting established forests deplete SOC, and (iii) indirect effects of climate change, such as wildfires, appear to have a greater impact on SOC than direct effects (e.g. rising temperatures). The outcomes from this research work are closely aligned the scope of the Journal of Nature Communications, and hence the manuscript is suitable for publication in this journal. However, the authors need to clarify following points before it can be accepted for publication.

- Line 54: May like to rephrase 'Soil organic matter (SOM) which comprised mainly of carbon is a key component of soils' (soil quality and soil health are regulated by SOM – not carbon perse).

Thank you for your comment, we have taken it into account in the new version of the text. The new sentence is now : 'Soil organic matter (SOM), mainly composed of carbon, is a critical component of soils¹ . '

- Line 60: May like to rephrase 'For this reason, decarbonizing soil has recently been ...mitigation. (<https://www.fao.org/3/ca6522en/CA6522EN.pdf>)

In the new version of the text, we have deleted this sentence from this paragraph, as the idea was already addressed in the introduction.

- Line 72 (may like to include this reference – (Ramesh, T., Bolan, N. S., Kirkham, M. B., Wijesekara, H., Kanchikerimath, M., Srinivasa Rao, C., . . . Freeman, O. W. (2019). Soil organic carbon dynamics: Impact of land use changes and management practices: A review. In D. L. Sparks (Ed.), *Advances in Agronomy* (Vol. 156, pp. 1-107). Cambridge, MA: Elsevier. doi:10.1016/bs.agron.2019.02.001)

Thanks for the reference, we have added it.

- Line 78-81. Crop residues and no-till farming are important land management practices which impact SOM (Chowdhury, S., Farrell, M., Butler, G., & Bolan, N. (2015). Assessing the effect of crop residue removal on soil organic carbon storage and microbial activity in a no-till cropping system. *Soil Use and Management*, 31(4), 450-460. doi:10.1111/sum.12215 Mehra, P., Baker, J., Sojka, R. E., Bolan, N., Desbiolles, J., Kirkham, M. B., . . . Gupta, R. (2018). A Review of Tillage Practices and Their Potential to Impact the Soil Carbon Dynamics. In D. L. Sparks (Ed.), *Advances in Agronomy* (Vol. 150, pp. 185-230). Cambridge, MA: Elsevier. doi:10.1016/bs.agron.2018.03.002)

Thank you for these interesting references, but we decided not to add them so as not to overload the list of references; and we already cite important papers on this subject.

- Line 107: Carbon Stock (Mg C/ha) – is it based on soil mass or soil volume

We have taken into account soil mass volume and soil mass method, as many meta-analyses mix the two indicators in their results and/or some do not specify this information. We have amended the text to be clearer on this point.

- Line 131: There are two types of forests – i.e., managed (agroforestry/plantation forest) and unmanaged forests (parkland/native); perhaps may like to make a statement on this; forest land management practices apply only to managed forests

Our manuscript is organized in different parts, the effect of land use change, management practices and climate change. Within the management/agricultural practices part, all practices studied are related to direct human intervention on the state of the environment, e.g. forest harvesting, establishment of species mixes. There is therefore little possible confusion.

- Line 138: Most wetlands are unmanaged; therefore, land management practices may not apply for wetlands – which is reflected in Figure 2 – no change in wetland land use (the last column in that figure)

We have reworked Figure 2 and the text to avoid confusion. However, our study can include wetlands, as there can be a change of land use from or to this type of environment. In addition, different human actions can take place on wetlands.

- Line 203: In relation to degraded land, one important land use change that has not been covered in this research work is mine-site revegetation which is increasingly used for carbon sequestration and carbon storage (Wijesekara, H., Bolan, N. S., Vithanage, M., Xu, Y., Mandal, S., Brown, S. L., . . . Surapaneni, A. (2016). Utilization of biowaste for mine spoil rehabilitation. In *Advances in Agronomy* (Vol. 138, pp. 292 pages). doi:10.1016/bs.agron.2016.03.001). Perhaps the authors may like to include a statement on the value of mine-site for carbon storage

Our study is based on land-use types as defined by the IPCC. The mine sites are certainly very interesting but are not part of the main IPCC land-use categories that we have used. Furthermore, we did not find meta-analytical results on this particular environment. For the sake of consistency, we have decided not to add sentences about these particular environments in our text, which would risk confusing our main messages.

- Line 245: Fertilizer input is a major contributor to soil carbon sequestration, especially in low fertile soils (Chowdhury, S., Bolan, N., Farrell, M., Sarkar, B., Sarker, J. R., Kirkham, M. B., . . . Kim, G. H. (2021). Role of cultural and nutrient management practices in carbon sequestration in agricultural soil. In *Advances in Agronomy* (Vol. 166, pp. 131-196). doi:10.1016/bs.agron.2020.10.001)

Thank you for this interesting reference. The new sentence is now: 'Fertilizer input is considered by some authors as a main contributor to soil carbon sequestration (up to 70–88 Mt

C yr^{-1} SOC increase globally estimated in Lessmann et al. (2022)³⁵), especially in low fertile soils⁵⁹. The reference 59 is Chowdhury et al., 2021

- Line 282: liming is an important land management practice to enhance soil carbon sequestration in acid soils (Pedro Luiz Oliveira de Almeida Machado¹, Vinicius de Melo Benites, Nanthi Bolan. 2021. 21. Liming Acidic Soils. RECARBONIZING GLOBAL SOILS A technical manual of recommended management practices. VOLUME 3: CROPLAND, GRASSLAND, INTEGRATED SYSTEMS AND FARMING APPROACHES PRACTICES OVERVIEW. Food and Agriculture Organization of the United Nations Rome, 2021, 289-303)

Our results include the effect of liming. These detailed results can be found in the tables associated with the figures. However, we had to make a choice among the 220 drivers analyzed, and we, therefore, concentrated on the most significant ones.

- Figure 4: Climate change can lead to increasing intensity and frequency of extreme weather events including drought, flood and bushfire. The authors have covered the effects of drought and bushfire on soil carbon; flooding can cause a decline in Soil carbon through leaching of dissolved organic carbon and erosion of sediments; perhaps the authors may like to include a statement on that.

We have amended the text to make more room for knowledge gaps. Thank you for the feedback. For example see the sentence “climate change can also have a significant impact on global SOC stocks, by increasing SOC mineralization due to higher temperature²⁸, or by decreasing carbon inputs to the soil as a result of less favorable plant growth conditions linked to more variable and extreme weather events²⁹” Or the sentence : “Other indirect effects of climate change, such as the impacts of flooding and the freeze/thaw effect, have yet received limited or no synthesis in existing meta-analyses.”

Reviewer #3 (Remarks to the Author):

Overall, an impressive study with highly relevant results. The authors used a second order meta-analysis and nicely show the distinct effect of either converting one land-uses to another or applying an improved management technique on an established land use. I think the strength of this study is to give global guidance on which conversions work and which managements are best practice. I see two caveats that result from the structure of the study, which I think should be highlighted in the discussion of results to avoid misinterpretation by the audience

1) through assessing relative change of SOC from one land use to the other, the results only apply when one land use is replaced by another and the basis of SOC is always the current level. It should, for example, be avoided that people get the impression that more carbon can be gained from afforestation of cropland than is lost from deforestation for cropland (Figure 2 could give that impression, if not looked at carefully).

Thank you for this comment, which is indeed important to clarify. We have added a sentence below Figure 2 to clarify this point. We also developed the idea in the text: "The high SOC

increase should therefore be interpreted with caution, as the initial level of SOC in cropland is low”.

2) The joint analysis of a global dataset is at the same time strength and weakness of the study. It gives a very nice overview on what global policy should focus on and at the same time masks small scale heterogeneity, which is still highly relevant for local implementation.

We agree with this comment. Our global analysis allows us to clearly classify the most promising practices to increase SOC globally or main threats to SOC. This type of results is particularly useful to set up international public policies (CAP...), funding programmes, etc. We also believe that this kind of result can be, in some circumstances, useful if combined with local knowledge to help local actors to make their decision. However, we agree that the local context must be taken into account in order to adapt each practice to the pedo-climatic (and social) context.

Nevertheless, to clarify this point we have stated our objective: "A high-level synthesis of existing knowledge can facilitate evidence-based decision-making and prioritize actions to globally increase SOC^{37,38}. Combined with local knowledge, it can also contribute to identifying the best practices for local implementation of SOC preservation and restoration."

The discussion would benefit if this issue would be addressed and if the authors would delineate more clearly for which purpose the study serves best (IMHO setting global priorities), and what it cannot tell with as much certainty (how it will play out at the local scale). I do think that the manuscript does need some medium revision, e.g. some parts of the methodology (exact inclusion/exclusion criteria, statistical model) are not expressed with sufficient detail,

We have reworked the material and method for more clarity. For example we listed the inclusion criteria as follows: 'To be included, a paper had to i) analyze the effect of one or several drivers on bulk SOC stock or concentration, ii) present a statistical formal analysis of at least two primary studies on SOC, iii) present indicators of precision of the effect-sizes (standard errors or confidence intervals). '

We also write our model equation and define each term, see equation 1.

In addition, the protocol, a description of the database, and a preliminary analysis of the database are made available to the reader (see Beillouin et al., 2021 and beillouin et al., 2022).

and I also found some numbers that are probably wrong in the text.

Thank you for reading this carefully. We have carefully reviewed the materials and methods and re-specified the workflow we used. We have also created and shared an R package that allows reproducing the analyses, and allows readers to understand in detail all the subtleties of the models used.

We have also reworked all the figures to identify errors that had crept into the manuscript despite our careful proofreading.

See my specific comments for details. However, after successfully addressing those comments, I think the study will be of great value and deserves to be published in nature communications. A final suggestion: The authors may think of making the data and R code of

this article available upon publication. It is a highly interesting dataset to the scientific community and would likely increase the visibility of the article!

Thank you for the suggestion. We developed an R code and the data are fully available. The study is now completely transparent. The protocol has been published with the data in a DataPaper. The data are freely accessible in a repository, and the metadata are described in the Datapaper (Beillouin et al., 2021). We have created an R package to make the analyses reproducible.

We have made a real effort to make our work fit into a FAIR methodology, with a level of requirement that we have (too) rarely encountered in environmental sciences.

Specific comments:

L65 – particularly essential in which way?

We have reworded the sentence which was not very clear.

L67 – Not sure what is the link between irrecoverable carbon emissions and negative emissions. Is l65-67 essential to your work or could you delete this?

We have simplified the sentence but kept the idea. The aim here is to highlight the importance of preserving carbon-rich ecosystems.

L70 -71 The percentages here somehow contradict with your introductory sentences that SOC is the most important terrestrial C storage

We have reformulated the sentence. The wording was not clear.

L77 – Would be good to mention the asymmetry here. SOC is lost much faster than it is rebuilt.

Thank you for this suggestion. We added the idea.

L79 – manure application

We changed the wording.

L81 – You mean that greater efforts may be needed in the future?

Yes, we have now also developed the idea in the discussion

L89 – it is not fully clear to me what is meant with this sentence. If some NET are country specific, can such higher order analyses not mask this? I agree that such higher order analyses are valuable to prioritize. At the same time, local implementation should be done on local best knowledge.

We agree that the aim of our study is to provide a global overview of the main drivers of SOC loss and of the effectiveness of agricultural practices in sequestering soil carbon (We have also clarified our objective sentence). Local pedo-climatic conditions and implementation methods can impact the effectiveness of these practices. As per your suggestion, we have provided our complete database so that interested authors can explore whether the effects of the practices have been detailed based on specific moderators and can interpret the results

more easily for their situations. However, the global effects reported in the main text provide valuable information as well. If the average effect (with confidence interval) of practice is very low, it is highly likely that the practice will have a weak effect in the vast majority of situations. Additionally, the global effects estimated can still be useful when combined with local information to better understand the local effects of specific agricultural practices. This is demonstrated effectively in a Bayesian framework, but it was not the focus of our article.

L94 – while I agree that the higher sampling size increases statistical power, I think that flattening out small scale variability (part of it true variability, part of it measurement error) does not only have benefits. You should also acknowledge that to some part this eliminates existing small scale variability. A powerful example how this can also lead to the wrong conclusion is given in Bradford et. al. (2021) <https://link.springer.com/10.1007/s10533-021-00789-5>

Bradford et al. focus on the use of ecosystem biogeochemical models, which are parametrized with data at a local scale, and then used at different scales, i.e. larger scales. This approach could be problematic if the variability of the observed phenomena is not well considered in the parametrization. To address this, they suggest quantifying conditional form and effect sizes to synthesize causal knowledge (as shown in Figure 3).

In our observational study database, we recorded the main types of agricultural practices, as well as sub-types and every moderator that impacted the mean effect size. However, synthesizing the effect of all these moderators was challenging and difficult to present in the main text, as each study focused on different types of moderators based on the possible different structures of models. Thus, we focused on estimating the mean effect size precisely in this study.

Furthermore, downscaling results based on global analysis and upscaling results from a local to global scale (as could be done in our case) do not pose the same problem. In our case, we gathered information to provide a precise mean global estimate. This estimate has usefulness for decision-makers and identifies global knowledge gaps. As mentioned before, if used properly, it also provides information on the range of performance reachable locally by a particular practice. But to avoid any misunderstanding of our results, we modify the text to clarify this point, see for example in the conclusion ‘ The implementation of our results in local contexts, however, must be carried out with care because of the particular pedo-climatic context and the lack of representativeness of our results in certain environments. ‘

L98 – Please give a definition of what is a driver for you.

We precised the definition of driver: (i.e. any human-induced intervention that directly or indirectly causes a change in SOC)

L104/105 – This sentence is not clear to me. I think you should give a very brief but exact description here of what model you use. I do not understand the sentence about frequentist of Bayesian, usually it must be one of these. Which one did you use?

We have reworded the method-related sentences to clarify. see for example : ‘To ensure the robustness of our conclusions, we compared the estimates obtained through frequentist versus Bayesian inference methods (suppl. 2), but we present in the main text only frequentist estimates.

L107 – Good that you included both, but how did you deal with the difference? Transfer one into the other? Did you consider that there is a difference in data quality i.e. stocks based on equivalent soil mass > SOC stocks > SOC contents?

The meta-analyses may or may not have the uniqueness of the SOC and the method that the primary studies used (e.g. soil depth, mass). It is therefore difficult for us to have a higher level of detail than the raw material we use. We have therefore included everything in our analyses (main text) and made analyses of the stability of our results by comparing where possible whether there were differences between SOC Stock and SOC concentration for example (see supplementary materials).

L114-126 – I think it is good that you start with an overview of your data. At the same time I think this paragraph should be shortened, as the only really interesting thing is that climate change impacts are smaller than LUC and MGMT. That the latter two have a large range as there is e.g. both afforestation and deforestation included is not surprising and does not require that much text, in my opinion.

We have tried to shorten this section. At the same time we have included a few sentences on the amount of data available and on knowledge gaps, as suggested by another reviewer. This information allows the reader to critically interpret our results.

L127 – A definition of “main land management practices” should be given.

We have reworded the sentence to avoid this confusion.

Figure 2 – Is there an error with the confidence intervals of grassland > forest? Median is -3%, while CI is +10 to +4%.

Thank you for spotting this error and we have corrected it.

L181/2 – Anthropized?

We have changed the term

193- interesting option instead of means?

Thank you for the suggestion, we have replaced “means” by “option”

L207/8 - This very important point should be highlighted even more, maybe with a numeric example with absolute.

Unfortunately our meta-analysis results do not allow us to have changes in absolute values or initial SOC values, as this is information was presented in the meta-analyses

L210 -) missing

Figure 3 – Could you rename energy crop, else it sounds like it includes biogas maize (which I guess it is not biogas maize)

We rename perennial energy crop for clarity.

L226 – two dots

L237 – you should also address that organic amendments are often already used. Hence higher amendments at scale are often not even possible as they are limited by NPP. Biochar, as much more stable may be the exception here.

we have developed this subject a little further in the main text: Yet the scarcity of biomass in some regions, or competition with forages in some sub-Saharan countries, or the lack of mature technology could hamper the development of biochar. Besides, possible adverse effects of biochar on soil properties and biodiversity should be considered⁶¹.

L239/40 – Not sure if this second order effect really adds to the discussion here.

L243/44 – Were you not talking about the effect of replacing mineral before? So why these two sentences here?

We have deleted this sentence which was confusing

L250-52 – I disagree here. NPP is high in many areas of SSA. So there should be enough feedstock for biochar. Additionally, it has been proven highly effective in improving soil fertility (Kätterer, et al; 2022; <https://doi.org/10.1007/s13593-022-00793-5>)

See previous comment.

L266 – highly significant meaning $p < 0.001$? This is the only time you use this term.

We have changed the term, which was not defined in our article

L274 – I suggest you add a “Yet,” before “The effect of mixed species . . .”

done

L278 – How is no-till farming significant if the CI is from -1.6 to 2? The 9.2% cannot be, some of the numbers must be wrong.

Thank you for spotting this error, which has been corrected.

L309 – For grassland, CO₂ enrichment actually had a significant positive effect, right?

Thank you for spotting this error, which has been corrected.

L310/11 – Maybe for that reason you should write in this section that you did not find significant effects instead of that there “were” none.

We now have a clearer picture of the amount of data associated with each driver, which makes it easier to nuance/interpret our results.

L313 – Contrasting?

Done

L318 – This makes a lot of sense! Very interesting.

L330 – I think it is worth to mention how they should best be interpreted. In my opinion they serve as an important overview on what works best. Yet at smaller scales, it may still be advisable to look for local studies when planning land use, as for example indicated by your outliers of Fig. 3.

See previous comments.

L373&380 – Duplication is an important point. Did you somehow check, whether the meta-analyses you used did not use the same studies. I assume if there is a huge overlap between two meta-analyses one should be excluded. Else the same data is ruminated many times. You have not specified how you handled overlap.

In our study, we listed all the experiments used in each meta-analysis, which allowed us to calculate an overlap rate precisely. We detailed this methodology in the materials and methods section. For even more clarity, we have modified the figures to directly display this redundancy rate on figures. But, to answer you more directly to your question, we consider the level of duplicates between each pair of meta-analyses and include this 'matrix' in the model to lower the weight of 'redundant' studies.

L381 – would be good to also state these quality criteria

To avoid making the text too lengthy, we only provided a broad description of the quality criteria in this publication. We refer readers to a recently published paper (same dataset) that provides a detailed description of these criteria.

L386 – I am confused by your statement that you used ratios, as your outputs are given in %change from the baseline.

We will now provide a more detailed description of our workflow. In fact, we calculated a logarithmic ratio to apply our models, and to meet the normality assumptions, we then back-transformed our results into percent changes for easier communication of our findings.

L399/400 – I do not fully get this hierarchical model. It would be good to describe this better with some numerical examples. What is the difference between φ and ε ? From your description, it seems that they are the same, e.g. you have a random effect for each effect in each meta-analyses, but how can you distinguish it from the residual error term if they have the same number of observations. At least from your description it seems like that is the case. We have made an effort to better explain each term in our model. In fact, our model refers to a 3-level meta-analytical model. This type of model has now been described in numerous works. We explicitly refer to this name so that the reader can make the connection.

L412 – I miss information on which software and statistical packages were used to implement the analysis. Also, I am still confused about the frequentist vs Bayesian statement from the introduction? What method did you finally use? I now saw it is written below, but still have the feeling it should come earlier.

Our R codes are freely available. We clearly mentioned the type of software and the packages used to analyze our results.

L415 – As you worked with transformed data – how did you backtransform the confidence intervals? As they are uneven, it looks like you did it correctly, but it does not hurt to state the details here.

We used the exponential function to transform the results from log to non-log scale. We used the formula $1 - \text{results} * 100$ to express the results in % change.

LI425ff – It is good that you tested the robustness and it would be even better if you share with the reader if your models were in fact robust. Same goes for the Bayesian vs frequentist, below.

We detailed these sensitivity analyses in the supplementary materials.

Reviewer #3 (Remarks to the Author):

Overall, I think the quality of the manuscript has improved considerably. The authors have done a good job in addressing the comments well and improving the text. I also like how far the authors take their effort to make the work widely accessible and reproducible. I think the article deserves to be published, but I have one final major concern. Figure 2 is, despite not being wrong in the numbers, in my opinion is really misleading and an invitation for misinterpretation. For conversions between forest and cropland / grassland and cropland, the proportional arrow is always much bigger for cropland to grassland/forest than in the other direction. In the first review round and responses this has been discussed to be an effect of different initial levels and is sufficiently addressed in the text. Yet, the figure gives the impression to me that losses of SOC are easily (over)compensated by just going back to forest/grassland. This is opposite of reality. To avoid this impression, I suggest to change the following: 1) do NOT make arrows proportional to the relative change – just make them all the same size (this proportionality makes really only sense if you would normalize them to the same baseline SOC content). 2) Give ranges for the SOC stocks for each land-use (e.g. Q25,Q50,Q75), so that it becomes evident from the figure that -0.14 x the median of grasslands is likely larger than $+0.28$ x the median of croplands (example of grassland to cropland and cropland to grassland) and maybe 3) explicitly state in the caption that higher number from cropland to grassland/forest does NOT mean that initial levels can easily be reached.

Point-by-point response to reviewer 3 :

Reviewer #3 (Remarks to the Author):

Overall, I think the quality of the manuscript has improved considerably. The authors have done a good job in addressing the comments well and improving the text. I also like how far the authors take their effort to make the work widely accessible and reproducible.

Thank you.

I think the article deserves to be published, but I have one final major concern. Figure 2 is, despite not being wrong in the numbers, in my opinion is really misleading and an invitation for misinterpretation. For conversions between forest and cropland / grassland and cropland, the proportional arrow is always much bigger for cropland to grassland/forest than in the other direction. In the first review round and responses this has been discussed to be an effect of different initial levels and is sufficiently addressed in the text. Yet, the figure gives the impression to me that losses of SOC are easily (over)compensated by just going back to forest/grassland. This is opposite of reality. To avoid this impression, I suggest to change the following: 1) do NOT make arrows proportional to the relative change – just make them all the same size (this proportionality makes really only sense if you would normalize them to the same baseline SOC content). 2) Give ranges for the SOC stocks for each land-use (e.g. Q25,Q50,Q75), so that it becomes evident from the figure that -0.14 x the median of grasslands is likely larger than $+0.28$ x the median of croplands (example of grassland to cropland and cropland to grassland) and maybe 3) explicitly state in the caption that higher number from cropland to grassland/forest does NOT mean that initial levels can easily be reached.

We have modified Figure 2 to make it more explicit. We have made the pictograms representing the land-use proportional to the mean SOC content (data source: FAO: only mean SOC stock.ha-1 are available). This way the reader can quickly identify the carbon stocks per hectare and can more easily interpret the calculated SOC change values. We decided to keep the arrows proportional to the SOC change, as putting all the arrows of the same size would have made the figure less readable, without solving the concern of reviewer 3. However, we have repeatedly, including in the Figure legend, taken care to draw the reader's attention to the fact that the flows are calculated in relation to the SOC level of the initial land-use, and that it may be complicated to return to an SOC level of the initial land-use. For example:

-In the Figure legend: *"Given that the mean initial SOC levels vary across different land-use types, the effect sizes expressed as percent change should be interpreted in relation to these initial levels."*

-In text: *"However, it is widely acknowledged that ecosystem restoration often fails to fully recover the functions of the undisturbed ecosystems^{52,53}, including soil carbon sequestration⁵⁴."*

-In text: *"It should be noted that a high percentage increase in SOC following the conversion of croplands to grasslands or to forests does not necessarily indicate that the SOC levels in natural grassland or forestland can be easily achieved."*

-In text: *"Our figures may, however, underestimate the actual SOC losses from the conversion of forest lands because most of the underlying primary studies - and thus most of the resulting meta-analyses - quantified these losses within a time frame ranging from a few years to a maximum of a few decades following conversion, whereas the time to reach a new SOC*

equilibrium after a land-use change is much longer (e.g. estimated to be about 80 years for the conversion of grassland to cropland⁴³)."